# Forward optic flow is prioritised in visual awareness independently of walking direction

**Paweł Motyka**[1]*, **Mert Akbal**[2,3], **Piotr Litwin**[1]

**1** Faculty of Psychology, University of Warsaw, Warsaw, Poland, **2** Department of Neurology, Max Planck Institute for Human Cognitive and Brain Sciences, Leipzig, Germany, **3** Academy of Fine Arts Saar, Saarbrücken, Germany

* pawel.motyka@psych.uw.edu.pl

**Data Availability Statement:** All data and code used for the analyses are available on GitHub at https://github.com/Pawel-Motyka/SMPVR.

**Funding:** This research was supported by the National Science Centre (Poland, https://www.ncn.

## Abstract

When two different images are presented separately to each eye, one experiences smooth transitions between them–a phenomenon called binocular rivalry. Previous studies have shown that exposure to signals from other senses can enhance the access of stimulation-congruent images to conscious perception. However, despite our ability to infer perceptual consequences from bodily movements, evidence that action can have an analogous influence on visual awareness is scarce and mainly limited to hand movements. Here, we investigated whether one's direction of locomotion affects perceptual access to optic flow patterns during binocular rivalry. Participants walked forwards and backwards on a treadmill while viewing highly-realistic visualisations of self-motion in a virtual environment. We hypothesised that visualisations congruent with walking direction would predominate in visual awareness over incongruent ones, and that this effect would increase with the precision of one's active proprioception. These predictions were not confirmed: optic flow consistent with forward locomotion was prioritised in visual awareness independently of walking direction and proprioceptive abilities. Our findings suggest the limited role of kinaesthetic-proprioceptive information in disambiguating visually perceived direction of self-motion and indicate that vision might be tuned to the (expanding) optic flow patterns prevalent in everyday life.

## Introduction

Senses do not operate as independent channels that passively reflect the state of the environment. Experience in one sensory modality is shaped also by signals from other senses and acquired knowledge of how such signals relate to each other. Integration of multisensory information often brings adaptive advantages such as increased accuracy of unimodal percepts [1–3] and perceptually-guided actions [4–6]. While navigating our environment, we tend to rely on vision, which, despite being a source of rich and precise information, is also prone to being influenced by other senses. The extent of these influences on vision can be assessed with methods that exploit the fact that visual experience may change independently of the physical properties of stimulation. A prominent example is binocular rivalry, a phenomenon which occurs when different stimuli are presented separately to each eye, causing a person to experience

gov.pl) under grant no. 2016/23/N/HS6/02920 (to P.M.). The funders had no role in study design, data collection and analysis, decision to publish, or preparation of the manuscript.

**Competing interests:** The authors have declared that no competing interests exist.

continuous alterations between one image and then the other [7]. As both images compete for perceptual dominance, exposure to stimulation from other sensory modalities can promote the visual awareness of a stimulation-congruent image. Such effects occur when there is semantic and structural congruence between multisensory signals (e.g., when sounds and images indicate the presence of the same object or are presented at the same frequency) and comprise influences from various sensory modalities such as audition [8–11], touch [11–14], olfaction [15,16], and interoception [17]. Using the similar method of continuous flash suppression, analogous effects have also been demonstrated for the vestibular system [18] and proprioception [19]. These findings show that conscious perception of ambiguous visual input appears to be biased towards its most likely interpretation, given the sum of multisensory information.

Aside from external sensory signals, our voluntary actions can directly inform us about what is likely to be seen. Through our experiences we learn how different bodily movements change incoming sensory signals and impact surrounding objects. Surprisingly though, despite tight sensorimotor coupling, the evidence that action can clarify the content of visual awareness remains inconclusive. The direction of hand movements (e.g., turning a knob) has been repeatedly shown to facilitate the perception of action-congruent variants of bistable motion displays [20–24]. A similar effect was found in one study [25] using binocular rivalry. In that experiment, participants were asked to perform smooth hand movements with a computer mouse only when one of the stimuli (a slowly rotating sphere) was exclusively visible. It was found that these movements lasted longer (i.e., the sphere was visible for a longer period of time) when the direction and speed of rotation were controlled by hand movements than when the subject had no motor control over the stimulus. However, more recent studies–using both binocular onset rivalry [23] and continuous flash suppression paradigms [26,27]–did not find increased access to awareness of stimuli whose rotation is congruent with manual actions. It is possible that these discrepancies are partly due to the use of different methods for presenting stimuli. In contrast to paradigms that involve interocular suppression (i.e., binocular rivalry, onset rivalry, and continuous flash suppression), elements of images presented in bistable displays (e.g., moving dots) remain perceptually accessible, which renders their apparent movement or configuration more susceptible to top-down and voluntary influences [23,28]. Among the suppression-based paradigms, action effects on visual awareness have been found only for binocular rivalry–in which subjects can keep track of whether or not the currently seen stimulus changes in congruence with their actions. This is not the case for paradigms in which the stimuli are presented only until they are detected (i.e., onset rivalry and continuous flash suppression). This suggests that conscious feedback regarding visuomotor congruence might be necessary to yield such effects, and thus here we chose binocular rivalry as a paradigm to investigate action influences on visual awareness.

For this purpose, instead of circumstantial associations between hand movements and their consequences observed on a 2D screen, we exploited the early-acquired and persistent association between direction of global bodily movement and concomitant changes in optic flow. While moving towards a target, the centre of one's visual field expands outward from a focus point, and, conversely, moving away leads to an inward contraction of visual peripheries [29,30]. Hence, in natural circumstances, the speed and direction of walking map one-to-one onto specific shifts in visual perspective. We aimed to examine whether optic flows that appear veridical while walking predominate in visual awareness over non-veridical ones. Notably, this question has previously been tackled in a study by Paris et al. [31], in which participants observed outward- and inward-moving dots while walking forwards and backwards. A series of neatly controlled experiments did not reveal any effects of walking direction on the perceptual availability of congruent and incongruent optic flow patterns. We approached this

question by using an experimental design differing in three critical aspects that, in our opinion, could have limited the ability of that study to find positive results. First, instead of presenting shifting dots, we used a more realistic three-dimensional environment to boost general feelings of presence and locomotion through space (although it should be noted that in a supplementary experiment by Paris et al., stimuli depicting a hallway with patterned flat surfaces were used [31, S1 Fig]). Second, in that study, participants walked along a circle (with a radius of ca. two meters) while being shown visual stimuli which depicted linear self-motion. Clearly, this would have decreased the overall congruence of visuomotor signals. Therefore, we used a treadmill so that the motion and the perceived optic flows were both linear. Lastly, and most importantly, in their study, the speed of the presented visual stimuli was matched directly to the average speed of walking, despite the fact that, in general, physically accurate optic flows presented via head-mounted displays appear much slower than the actual walking velocities [32–36] (mainly due to the restricted scope of peripheral vision, which is a major contributor to sense of speed). Thus, in the current research, optic flow speeds were subjectively matched to the velocity of walking.

Furthermore, we aimed to examine whether the impact of bodily actions on vision is greater for individuals with more precise proprioception–the sense of self-movement and of the position of body parts in space. In general, our hypotheses follow from Bayesian models of multisensory integration, according to which inferences about the state of the environment–and thus our perceptual experience–rely on the degree of congruence between unimodal signals (here: visual and kinaesthetic-proprioceptive) and their relative precisions (understood as inverse variance/noisiness) [37–39]. The basic idea is that the extent to which one is aware of one's movement and body parts should reflect the overall reliability of action-related afferent signals. Hence, reliable bodily signals should more efficiently bias interpretation of an ambiguous visual scene towards an intermodally coherent percept. Whereas, in general, there is substantial evidence showing that the relative impact of sensory information increases with its precision [37,40,41], this is rarely controlled for in studies on crossmodal influences on visual awareness. This might be particularly important for detecting elusive bodily influences in this domain. Notably, in a study investigating cardiac effects on binocular rivalry [17], such influences were found only for participants who were better at detecting their own heartbeats. Like interoception, proprioception can be considered to be a heterogeneous modality, as it comprises processing of information from functionally distinct sensors such as joint and skin mechanoreceptors, tendon organs, and muscle spindles [42,43]. This could explain why performances on different types of proprioceptive tasks (e.g., active and passive movements) tend not to correlate [44] and might be variously associated with other sensory processes. Therefore, given the use of active, self-induced bodily movement in the current research, we decided to employ a task based on active reproduction of the position of one's limbs to assess interindividual differences in proprioceptive accuracy and noisiness of proprioceptive signals.

In summary, this study aimed to test whether the direction of one's locomotion can bias perceptual awareness for highly-realistic visualisations of self-motion in space. We hypothesised that optic flows congruent with walking direction would predominate in visual awareness over incongruent ones, and that this effect would be more pronounced for individuals with greater reliability of active proprioception.

## Method

### Participants

All participants had normal or corrected-to-normal visual acuity and no history of colourblindness, amblyopia, neurological or psychiatric disorders, or propensity to motion-sickness.

Eleven participants were included in a pre-study (5 females, mean age = 29.0 ± 4.5, range: 20–35 years) conducted in order to estimate optic flow speeds for the main experiment. The main sample comprised forty-one participants (23 females, mean age = 22.7 ± 2.8, range: 18–36 years), which was greater than the sample sizes previously used in similar experiments [17,23,25,31] by at least 35%, in order to enable the detection of potentially smaller effects and to compensate for the possible exclusion of any data. Data obtained from three participants were removed from binocular rivalry analyses as these participants did not complete all of the blocks due to technical issues. Additionally, nine participants were excluded as outliers (for criteria see section *Exclusions* below), so twenty nine participants were included in the final analyses (16 females, mean age = 22.7 ± 2.9, range: 18–30 years). Participants were recruited via social media. All participants gave written informed consent before taking part in the study and were compensated financially for their participation. The procedure was approved by the Ethics committee of the Faculty of Psychology of the University of Warsaw. All research was performed in accordance with relevant guidelines and regulations.

## Materials and apparatus

The proprioception evaluation system (Propriometr R [45]) was used to assess interindividual differences in active proprioception. This system includes an arm-mounted electric goniometer which allows continuous measurement of the deviation angle with an accuracy of 0.1˚, a remote button box held by participants, and dedicated software to record and store data. An electric rehabilitation treadmill (Insportline Neblin) with an additional safety belt was used to manipulate walking direction. All visual stimuli and instructions were presented using a head-mounted VR display (HTC Vive, HTC, Taiwan; refresh rate: 90 Hz; resolution: 1080×1200 per eye). Participants used the handheld controller to provide responses throughout the experiment by pressing the left or right part of the touchpad (marked with additional stickers; hereafter referred to as "buttons"). Button presses were sampled at 40 Hz. The program and the VR environment were created using Unity version 2018.2.11f1 (Unity Technologies, San Francisco, California). A wide default field of view on the HTC Vive headset (110˚ for both horizontal and vertical axes) was used to preserve feelings of presence and speed in the virtual scene. The scene depicted a flat linear path in a 638 m long, 1.80 m wide, and 2.30 m high tunnel with cave-like walls and ceiling, and was chosen from a set of pre-tested designs in order to evoke a relatively low level of mixed percepts despite its wide field of view. The "bumpy" structure of the environment provided peripheral cues contributing to feelings of self-motion, but, at the same time, its uniformity as it recedes into the distances gives no information about distance travelled and did not convey conflicting (non-motion related) cues to different eyes. Head movements were accurately mapped in the virtual scene, which was presented either as green (RGB: 0, 140, 70) or red (RGB: 140, 0, 70), with shades slightly varying due to the use of natural shadows in the environment. The same virtual scene was used both in the pre-study determining the optimal optic flow speeds and in the main experiment.

## Determination of optimal optic flow speeds in relation to walking velocity

In this pre-study, participants walked either forwards or backwards while viewing the same visualisation of self-motion through the virtual tunnel with both eyes (the direction of visual movement was always consistent with the walking direction). A constant walking speed of 0.42 m/s (1.5 km/h) was used throughout the experiments. This value was chosen because it was identified in the pilot phase as the maximum velocity at which one can comfortably walk backwards (it's worth noting that locomotion on a treadmill is perceived as being faster than equally fast locomotion on the ground [46]). The participants' task was to adjust the optic flow

speed to match the pace of walking as closely as possible by pressing either the left or right button to slow down or speed up, respectively, the visually perceived speed (with a minimum step of 0.002 m/s change). Participants were instructed to take as much time as they needed to find the subjectively veridical speed and to cease further adjustments once they believed that "it is moving as fast as it should–neither too fast nor too slow". Participants provided estimates in eight different conditions (presented in a pseudo-randomised order), exhausting all combinations of the following factors: walking direction (forward/backward), visualisation colour (red/green), and starting speed (unrealistically slow [0.10 m/s] or unrealistically fast [2.00 m/s] in relation to the pace of walking). Before the actual task, participants familiarised themselves with the VR environment and walking on a treadmill, and were instructed to focus their gaze on a point located at about 2/3rds of the way down the tunnel. Median values of matched optic flow speeds were calculated separately for both walking conditions for each individual. Then, medians were calculated for the whole sample and set as fixed optic flow speeds for the respective types of optic flow (i.e., expanding and contracting) in the main study.

### Proprioception assessment

Interindividual differences in proprioceptive abilities were assessed using the active joint position reproduction task [43,45]. Blindfolded and seated participants were asked to reproduce positions of their dominant arm as accurately as possible. The procedure included two types of movement (flexion and abduction at the glenohumeral joint) and three target positions (60˚, 90˚, and 120˚ deviations from a vertical axis; Fig 1A). Each trial followed the same structure: 1)

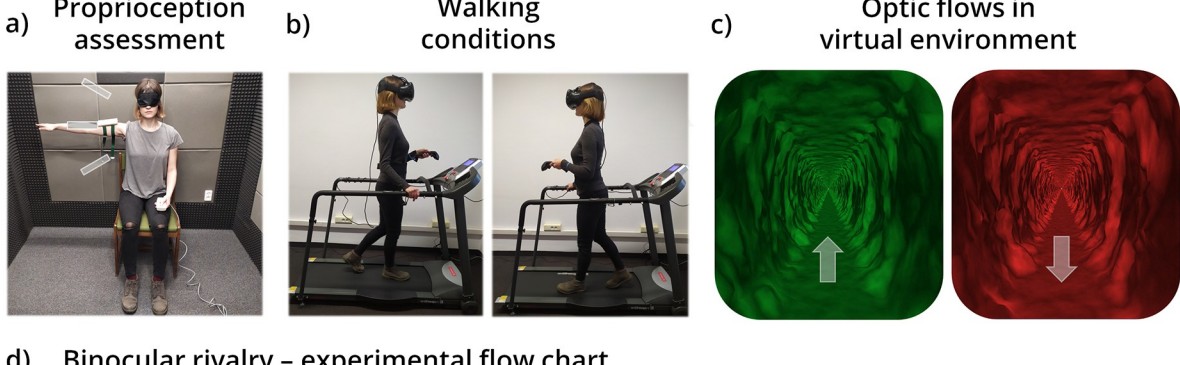

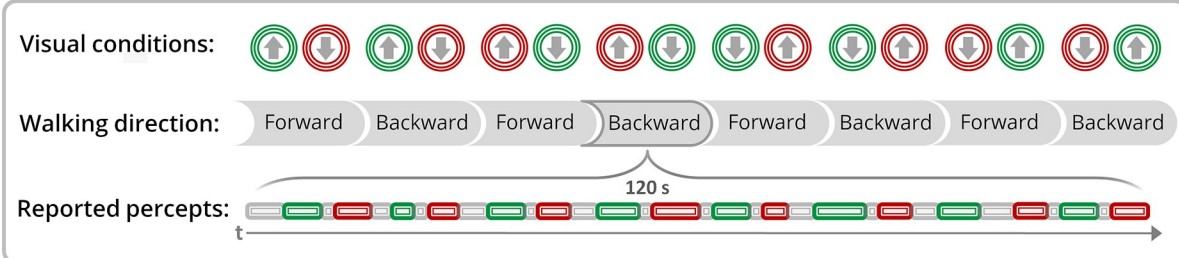

**Fig 1. Experimental procedure.** (**a**) Interindividual differences in active proprioception were assessed with a joint position reproduction task (the picture shows flexion at 90˚). (**b**) During the main task, participants walked either forwards or backwards on a treadmill while being presented with (**c**) different optic flow patterns (expanding and contracting) to each eye via the head-mounted display. Their task was to continuously report with a hand-held controller which of the coloured visualisations (red or green) was seen at each moment. (**d**) The experiment comprised eight blocks counterbalanced with respect to walking direction as well as the direction and colour of visualisations seen by each eye. (The brightness and contrast of the images have been adjusted for illustrative purposes; we obtained informed consent to publish the photographs).

after an experimenter placed the participant's arm in the target position, the participant was asked to stabilise their arm and to press a button on a device held in their non-dominant hand to record the exact position; 2) next, the participant's arm was slowly pulled down to the default position (0˚) by the experimenter; 3) finally, the participant had to reproduce the target position and confirm it by pressing the button. During the training phase, participants were instructed to perform the task in an attentive and measured manner, but discouraged from spending more than ten seconds calibrating the position of their limb. The assessment lasted about twenty minutes and consisted of thirty reproductions in a fixed order, starting with fifteen abductions at interchangeable positions (60˚, 90˚, 120˚), followed by an analogous series of flexions. Proprioceptive accuracy was operationalised as the inverse of the mean difference between target and reproduced positions (error) and proprioceptive precision as the inverse of the variance of the distribution of errors.

## Binocular rivalry between optic flow patterns when walking forwards/backwards

In the main task, participants walked forwards and backwards on the treadmill (Fig 1B) while viewing visualisations of self-motion through a virtual tunnel (Fig 1C), with one eye seeing motion consistent with walking forwards (i.e., expanding optic flow) and another consistent with walking backwards (i.e., contracting optic flow). Their task was to continuously report (by holding down either the left or right button) which of the presented coloured visualisations (red or green) was seen at a given moment. Participants were asked to refrain from pressing any button when perceiving mixed visualisations. The assignment of particular buttons to colors was counterbalanced between participants. Participants reported the colour, rather than optic flow direction, for two reasons: the pilot phase indicated that it was more straightforward to report, and, more importantly, use of a quality orthogonal to the quality of interest alleviates potential response bias (see here [12,23] for a similar approach).

Participants completed eight 120-second blocks (presented in a pseudo-randomised order), exhausting all combinations of the following factors: walking direction (forward/backward), direction of optic flow (expanding/contracting) seen by each eye, and the colour of visualisation (red/green) seen by each eye (Fig 1D). Participants were encouraged to take breaks between blocks to help maintain focus throughout the experiment. Individual blocks were repeated if participants had any problems or reported mistakes pressing the buttons. To prevent such issues, participants were initially familiarised with the VR environment and practiced walking on a treadmill without a headset. Experimental blocks were also preceded by four training blocks: two 45-second standing blocks and then two 90-second blocks with different walking directions (the colours and directions of optic flows presented to each eye were pseudo-randomised). The training phase was extended if participants still experienced any problems with the task. As was done when determining the optimal optic flow speeds in the pre-study, participants were instructed to focus their gaze about 2/3rds of the way down the tunnel throughout the experiment.

After the completion of all experimental blocks, participants were asked to rate their confidence that the button presses had overlapped in time with the dominant visual percepts they had experienced. Answers were given on a 7-point Likert-type scale with the following extreme statements: (1) "while pressing a button I was *never* sure whether I was properly indicating the dominant visualisation" and (7) "while pressing a button I was *always* sure that I was properly indicating the dominant visualisation". Post-experiment interviews found that none of the participants had identified the goal of the experiment and that most of the participants were not aware that each eye had been presented with a different image.

## Questionnaire measures

Upon arrival, participants completed a shortened version of the Simulator Sickness Questionnaire [47] (SSQ), which they also filled-in at the end of the experimental session. The questionnaire contained seven items assessing the severity of symptoms that may occur after a VR experience (included in S1 Table). The difference in means between post- and pre-test served as a control measure of cybersickness induced by the procedure. Upon completion of the experiment, participants also answered an adapted version of the Slater-Usoh-Steed Questionnaire [48] (SUS), which measures the sense of presence in a virtual scene (S2 Table). Our aim was to control for the induced level of immersion and to explore its potential association with the predominance of congruent optic flows during binocular rivalry.

## Statistical analysis

**Preprocessing of binocular rivalry data.** Button presses were logged throughout the experiment at a frequency of 40 Hz. The data obtained were then processed to identify instances of left and right button presses (exclusive percepts) and periods when no button was being pressed (mixed percepts). For each subject, individual intervals of particular response types (in milliseconds) were aggregated separately for different walking conditions, and their cumulative dominance (i.e., percentage share of periods of expanding, contracting, and mixed optic flows) was used as a measure of perceptual awareness in the main analysis. For the exploratory analysis of time-course changes in perceptual dominance throughout the average block, the proportion of particular responses was calculated for consecutive one-second windows of blocks of walking forwards and backwards. This complementary analysis allowed a more fine-grained examination of perceptual dominance patterns unfolding over the time-course of binocular rivalry. Finally, perceptual alterations were defined as perceptual switches from one image to another (e.g., left–none–right button presses), excluding return transitions (e.g., left–none–left).

**Exclusions.** Before the main analysis, we attempted to exclude data from participants who did not experience or properly report percepts during binocular rivalry. Potential outliers were identified based on three dimensions: a) mean number of alterations per block, b) mean duration of exclusive percepts, and c) cumulative duration of mixed percepts. A cut-off criterion of ±3 SD was used for each case. Based on these criteria, one subject was excluded due to excessively long mean duration of exclusive percepts (9.63 s; SD = 4.25). However, closer inspection of the data revealed a discernible cluster of seven subjects with very low mean duration of left/right button presses (< 0.32 s; the next lowest value was 0.86 s), and correspondingly, a very large cumulative proportion of mixed percepts (> 90%, with the next highest being 74%). These seven subjects were excluded from further analyses. In addition, one participant with virtually no alterations (M = 0.25) was excluded. While some of these outlying response patterns might be due to strong eye dominance, it could be also that some of the participants did not accurately follow the instruction to hold down the button as long as visualisation of a given colour dominated, instead making short clicks, which could go unnoticed by the experimenters. None of the remaining subjects were excluded based on low self-reported degree of overlap between button presses and dominant percepts (with answers below 3 on the 7-point scale as the cut-off criterion). In total, 9 out of 38 subjects were excluded from the binocular rivalry analyses (for results from the unfiltered sample, see Supplementary Information). Only behavioral results for the proprioceptive task were based on data without any exclusions.

**Statistical testing.** To test the hypothesis that locomotion-congruent optic flows would predominate in visual awareness over incongruent ones, a two-way repeated measures analysis of variance (ANOVA) was used, with optic flow congruence (congruent/incongruent) and

walking direction (forward/backward) as factors, and cumulative periods of dominance of particular percepts (in %) as the dependent variable. We expected to observe a main effect for congruence and no interaction. Pearson correlations were used to examine the hypothesis that the effects of bodily movement on visual awareness would increase with greater precision of active proprioception. We anticipated that the degree of predominance of locomotion-congruent percepts (i.e., difference between congruent and incongruent dominance durations) would decrease with nosier proprioception (indexed by the variance of errors) in both walking conditions. Equivalent non-parametric tests were used when appropriate. A two-sided alpha level of 0.05 was used in all tests. The statistical analyses were conducted using R version 3.5.1 [49] with RStudio version 1.1.463 [50]. All data and code used for the main and supplementary analyses are available on GitHub at https://github.com/Pawel-Motyka/SMPVR.

## Study 1: Results and discussion

### Determination of optimal optic flow speeds in relation to walking velocity

The matched speed of optic flow for walking forwards was significantly higher (M = 0.95, SD = 0.31 m/s) than for walking backwards (M = 0.69, SD = 0.23 m/s, t(10) = 5.61, p < 0.001, Cohen's d = 1.69; S1 Fig). These results are in line with previous findings about speed perception in virtual environments [32–36], in that they indicate a sizeable overestimation of matched optic flow speeds in relation to actual walking velocity (0.42 m/s); however, this is a novel demonstration that this tendency is more pronounced for the more-familiar forward locomotion. The overall medians from the forward (Me = 0.95 m/s) and backward (Me = 0.60 m/s) conditions in this sample were set as speeds for respective types of optic flow (i.e., expanding/contracting) in the main study.

Further exploratory comparisons were run to control for the role of colour and starting speed. There was no significant difference in matched optic flow speeds between red (M = 0.81, SD = 0.25 m/s) and green visualisations (M = 0.84, SD = 0.27 m/s, t(10) = −1.64, p = 0.132), whereas the starting optic flow speed was shown to significantly bias the estimations–they were higher for the unrealistically fast starting speed (M = 0.89, SD = 0.30 m/s) compared to the unrealistically slow one (M = 0.76, SD = 0.23 m/s, t(10) = 2.89, p = 0.016, Cohen's d = 0.87). This effect is not surprising given the well-documented robustness and ubiquity of the anchoring effect [51]–the tendency to be influenced by a reference point presented prior to the decision-making process.

### Effects of locomotion on perceptual awareness of optic flow patterns

Contrary to our hypothesis, there was no main effect of congruence (F(1, 28) = 0.01, p = 0.943, $\eta^2_G < 0.001$) on the cumulative durations of percepts, the main effect of walking was above the threshold of significance (F(1, 28) = 4.03, p = 0.054, $\eta^2_G = 0.006$), and there was a significant interaction between both factors (F(1, 28) = 52.1, p < 0.001, $\eta^2_G = 0.130$; Fig 2A). Post hoc Bonferroni-corrected comparisons showed that, for walking forwards, congruent optic flow was perceived for longer (M = 36.7%, SD = 11.1%) than incongruent flow (M = 29.6%, SD = 8.13%, t(28) = 5.74, p < 0.001), whereas for walking backwards, incongruent optic flow (M = 38.2%, SD = 9.42%) predominated over congruent flow (M = 31.0%, SD = 8.78%, t(28) = 5.80, p < 0.001). This indicates that expanding optic flow (naturally coupled with forward locomotion) was more likely to gain access to visual awareness than contracting flow (consistent with backward self-motion), independently of walking direction. The dominance durations of both expanding (t(28) = 1.47, p = 0.885) and contracting (t(28) = 1.37, p = 1.000) optic flows did not differ significantly between walking conditions. The mean duration of mixed percepts (i.e., proportion of time without any button presses) showed a non-significant

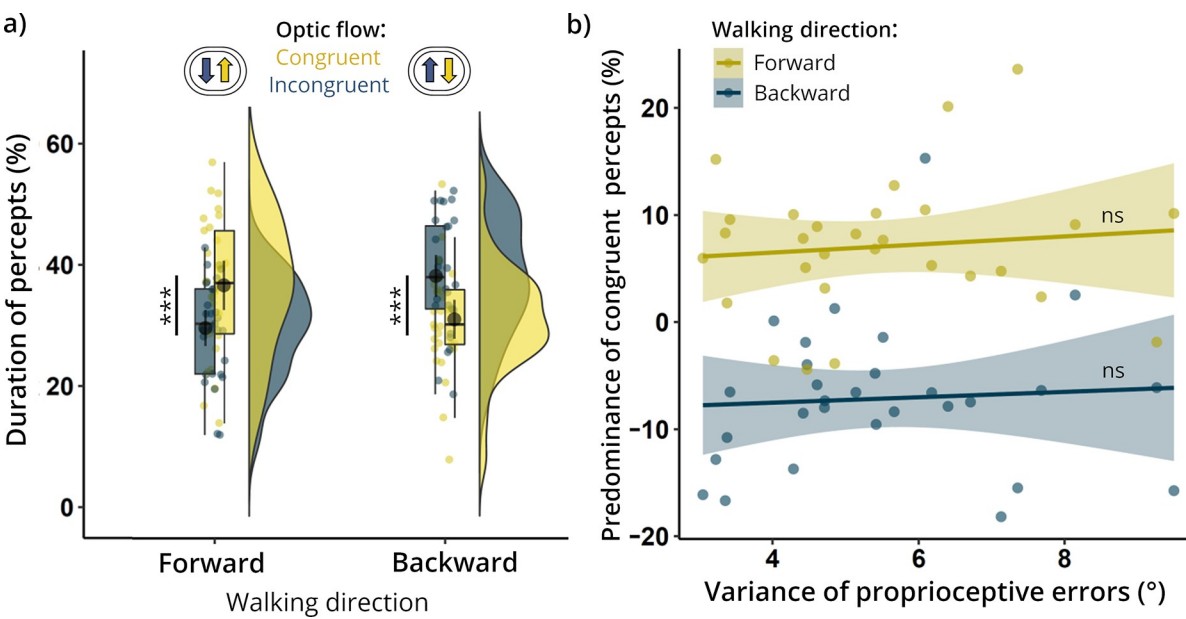

**Fig 2. Effects of locomotion on perceptual awareness of optic flow patterns.** (**a**) Cumulative durations of perceived congruent and incongruent percepts for each walking direction. Expanding optic flow (congruent with forward locomotion and incongruent with backward movement) dominated visual awareness independently of walking direction. (**b**) Relation between propensity to observe locomotion-congruent percepts (y-axis) and noisiness of active proprioception (x-axis). The expected negative correlations were not observed for either forward or backward locomotion. *** $p < 0.001$; ns = non-significant.

tendency to be higher when walking forwards (M = 33.7%, SD = 18.4%) than when walking backwards (30.7%, SD = 16.8%, t(28) = 2.01, p = 0.054). Additionally, there was no significant difference in mean number of perceptual alterations per block between forward (M = 26.1, SD = 11.1) and backward (M = 27.1, SD = 10.6, t(28) = −0.72, p = 0.477) conditions. The self-reported level of overlap between button presses and dominant percepts was relatively high (M = 5.14, SD = 0.79, range: 3–6, on a 7-point scale). The results from the unfiltered sample match closely those reported above and are included in the Supplementary Information (S2 Fig).

## Relation between proprioceptive precision and the predominance of congruent optic flows

In the proprioception assessment, the mean error on the active joint position reproduction task–the (inverse) index of proprioceptive accuracy–was 5.41˚ (SD = 1.84˚, range: 2.20˚–9.02˚), whereas the mean variance of the error distribution–the (inverse) index of proprioceptive precision–was 5.49˚ (SD = 1.81˚, range: 3.05˚–10.30˚). There was a positive correlation between performances for abductions and flexions, both in terms of proprioceptive accuracy (r(39) = 0.49, p = 0.001, S3A Fig) and proprioceptive precision ($r_s$(39) = 0.54, p < 0.001, S3B Fig). As proprioceptive accuracy and precision were found to be strongly correlated ($r_s$(39) = 0.79, p < 0.001, S3C Fig), only proprioceptive precision was used in further analyses (as it is directly relevant to our hypothesis). Contrary to our predictions, we found no evidence of any association between proprioceptive precision and the difference between congruent and incongruent optic flow durations (in terms of percentage points), either in the forward (r(27) = 0.10, p = 0.593) or backward ($r_s$(27) = 0.15, p = 0.427) walking conditions (Fig 2B).

## Self-reported levels of simulator sickness and sense of presence

Symptoms of simulator sickness (SSQ questionnaire) were more likely to appear after VR-exposure than at the beginning of the experiment ($V = 78$, $p = 0.013$, Cohen's $d = 0.48$), although they were rarely observed in both cases (before: $M = 1.12$, $SD = 0.17$, range: 1.00–1.71; after: $M = 1.26$, $SD = 0.28$, range: 1.00–2.14; 1–4 point severity scale with 1 indicating "none"). Self-reported sense of presence in the virtual–and perceptually ambiguous–scene (SUS questionnaire score) was moderate ($M = 3.59$, $SD = 1.09$, range: 1.33–5.50; 1–7 point scale). All intercorrelations between questionnaire items (Q1-Q6) were positive, and one third of them were significant (S4 Fig). Sense of presence was not associated with the frequency of perceptual alterations ($r(27) = 0.05$, $p = 0.786$), but it was found to decrease as the proportion of mixed percepts increased ($r(27) = -0.56$, $p = 0.001$, S5A Fig). In addition, sense of presence showed a non-significant tendency to increase with the degree of predominance of congruent percepts ($r(27) = 0.35$, $p = 0.064$, S5B Fig). These results suggest that the highly ambiguous (perceptually unstable) environment, and possibly also the violation of our life-long experience of optic flows being congruent with direction of motion, might undermine one's sense of presence in the virtual scene.

## Time-course changes in perceptual awareness of optic flow patterns

Here we aimed to uncover the dynamics of perceptual changes throughout the blocks. This exploratory analysis assessed whether the probability of perception of particular stimuli (e.g., locomotion-congruent optic flows) shifted in time during exposure to a visually ambiguous environment. For example, it could be that the prevalence of congruent percepts built slowly with accumulating evidence of movement in a given direction. Alternatively, in-line with a recent theoretical model [52], perception might have been initially biased towards prediction-consistent percepts and then subsequently shifted to reflect more surprising (informative) events. Such effects have been reported for timescales of hundreds of milliseconds, but it has been suggested that they may generalise to longer timescales [52]. To examine the possibility of such non-linear associations between the progress of time and the predominance of particular percepts, a generalised additive model [53,54] (GAM) was used to fit the data. First, we calculated the relative proportions of percepts (in %) in consecutive one-second windows of each individual block (1–120 s). Next, the proportions were averaged over different time windows separately for forwards- and backwards-walking blocks and each subject. A regression model with cubic splines was used to estimate the variations in probability of perceiving congruent, incongruent, and mixed optic flows over the course of an average block (with a generalised cross-validation smoothing parameter and $k = 10$). In both walking conditions, the level of deviance explained was comparably modest (forwards: 3.06%; Adj.$R^2 = 0.028$; backwards: 3.93%; Adj.$R^2$ of 0.037) and all smooth terms were significant, indicating the presence of block-wise variations in awareness of congruent, incongruent, and mixed percepts (all p values < 0.001, except for congruent percepts when walking forwards: $p = 0.031$). The results indicate that, after an initial phase without button presses, the predominance of expanding optic flow emerged and remained relatively stable over time for both forward and backward locomotion (Fig 3A). Notably though, mixed percepts increased gradually while walking forwards (alongside an associated decrease in incongruent percepts), which was not observed for walking backwards (Fig 3B). In summary, these exploratory results provided no support for any of the supposed time-dependent effects of locomotion on perceptual awareness.

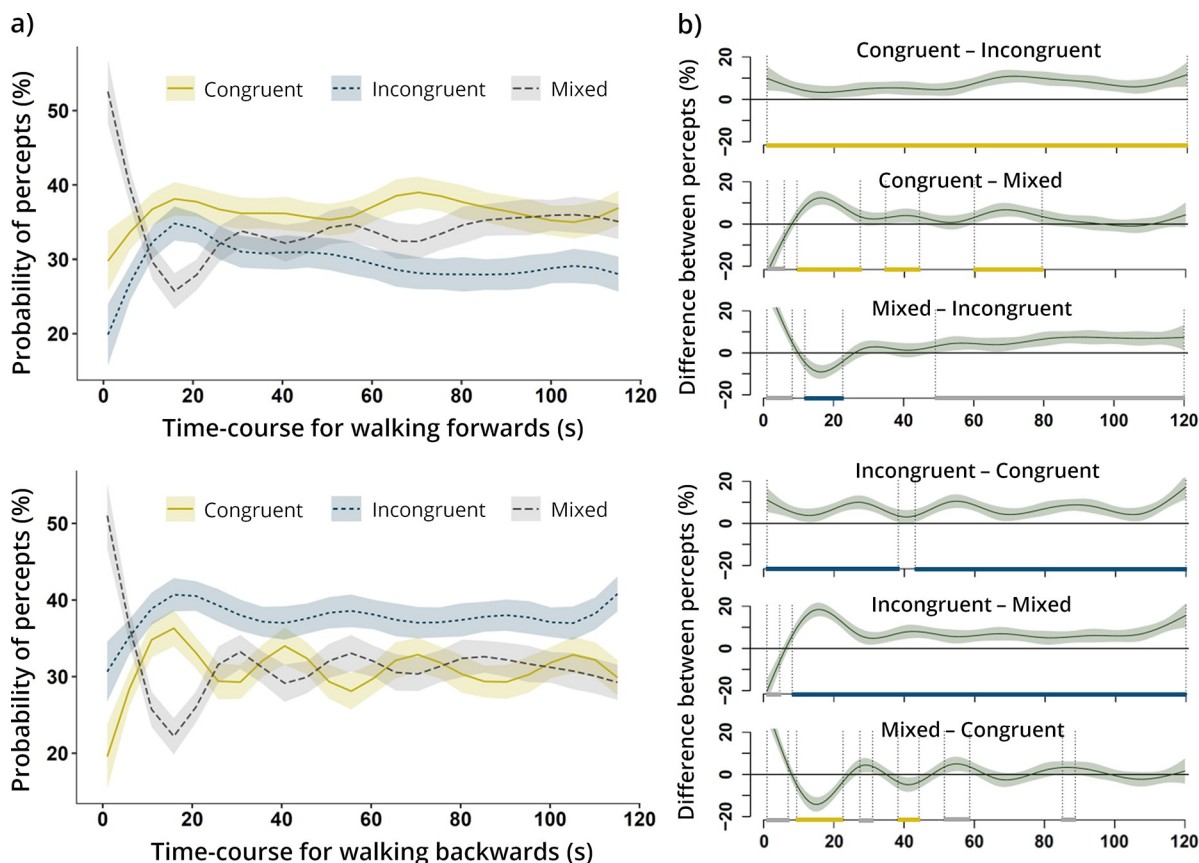

**Fig 3. Time-course changes in perceptual awareness of optic flow patterns.** (**a**) Non-linear smooths (fitted values) representing the probability of perceiving congruent, incongruent, and mixed percepts over the time of average blocks. The pointwise 95%-confidence intervals are depicted by the shaded bands. (**b**) Pairwise contrasts between probabilities of different percepts. Time intervals during which one of the compared percepts was significantly more likely to be perceived (i.e., shaded confidence bands do not overlap with zero) are bracketed by vertical dotted lines and marked by coloured segments on the x-axis (with the colours indicating the predominant percept).

## Comparison of expanding optic flow dominance while standing and walking forwards/backwards–supplementary analysis

The fact that, instead of the hypothesised congruence effects, we observed perceptual prioritisation of expanding visual flow in both walking conditions might raise the question of whether these results were merely due to the intrinsic visual bias or rather to a combination of the bias and the influences of self-motion (we thank an anonymous reviewer for bringing this point to our attention). To at least partially address this question with the available data, we compared dominance durations of expanding optic flow between two training blocks (in which participants stood still) and experimental blocks with equivalent combinations of visual factors. As each combination occurred once in both walking conditions, particular training blocks could always be matched with their forward and backward self-motion counterparts. Additionally, the duration of these paired blocks was shortened to the length of the training blocks (so as to level out the relative significance of the initial preponderance of mixed percepts; cf. Fig 3A). Exploratory pairwise comparisons were run to examine differences in dominance durations of expanding optic flow between a) standing and walking forwards, and b) standing and walking backwards (n = 28, due to missing training data from one subject).

The dominance of expanding optic flow while standing (M = 33.2%, SD = 12.2%) was not significantly different from its dominance when walking forwards (M = 34.8%, SD = 12.4%, t(27) = −0.55, p = 0.587) or backwards (M = 37.2%, SD = 12.7%, t(27) = −1.46, p = 0.157; S6 Fig). Furthermore, to assess whether these non-significant results reflected genuine null effects rather than insufficient statistical power, the corresponding Bayes factor analyses were performed (using the default Jeffreys–Zellner–Siow prior, as we lacked justification for informed prior specification). As for the comparison between standing and walking forwards, there was substantial evidence for the null hypothesis ($BF_{10}$ = 0.23). However, in the case of comparison between standing and walking backwards, we found only anecdotal support for the null effect ($BF_{10}$ = 0.51). These analyses suggest that forward self-motion does not enhance perceptual awareness of expanding optic flow relative to the stationary position. Similar support for the null effect cannot be fully concluded for the case of walking backwards. Nevertheless, altogether, these exploratory results do not yield indications that the primary effects of this study could have been caused by a specific/coarser form of visuo-motor interactions (e.g., increased predominance of a more familiar visual flow with any type of self-motion).

## Study 2

The results of the main study did not support our initial hypotheses. First, we did not find evidence for increased perceptual awareness of optic flow patterns congruent with direction of walking. Additionally, there was no association between precision of active proprioception and the propensity to perceive locomotion-congruent optic flow patterns. Instead, we found an overall perceptual preference for expanding optic flow (consistent with forward movement) over contracting flow. It is worth noting that optic flow speeds were chosen based on results from our pre-study, which found that when forward and backward optic flows (expanding and contracting, respectively) appear subjectively matched to a given walking pace, the forward flow will be faster than the backward one. Given that the main hypothesis focused specifically on the perceptual availability of expanding and contracting optic flow patterns in different walking conditions, we prioritised the veridicality of optic flows relative to walking pace over having equal optic flow speeds. Thus, the overall predominance of expanding optic flow could have stemmed from its higher velocity rather than a perceptual preference for the particular direction of the optic flow. Therefore, we designed a follow-up study to clarify whether this effect was merely due to the difference in the speed of visualisations.

We reasoned that if it was only speed of optic flows which determined the effect observed in the previous study, we would observe the opposite pattern of results if we were to switch the speeds of the backward and forward optic flows. Working from assumption that the speed was the sole cause of the observed differences, we calculated the sample size that would be required to reveal a perceptual preference for the faster (i.e., backward) optic flow. To this end, we calculated 90% confidence intervals for the effect size from the main study, and determined the sample size sufficient to detect (with 99% statistical power) an effect size equal to the lower bound value of the interval [55]. Significant perceptual preference for backward optic flow would mean that the direction of optic flow was not responsible for the differences observed in the main study. On the contrary, if neither optic flow predominated, that would indicate that the direction of optic flow is an independent factor determining access to visual awareness (cancelling out the speed-based effect). Finally, if the pattern of results from the main study holds despite the inversion of optic flow speeds, it would mean that the direction of optic flow is the sole cause of observed differences. In summary, the current study was designed in a falsificatory manner to either exclude or support the explanation that the previously-observed effects resulted solely from the difference in optic flow speeds.

The experimental procedure mirrored the one used in the main study, with two exceptions: a) the speed of the backward optic flow was set to 0.95 m/s (3.42 km/h), while the forward optic flow was 0.60 m/s (2.16 km/h); and b) proprioceptive assessment as well as SSQ and SUS questionnaires were skipped. The required sample size (9 subjects) was augmented to compensate for possible exclusions, which amounted to about 25% of participants in the main study. Hence, a group of twelve new subjects was recruited for the current study (6 female, mean age = 23.3 ± 2.4, range: 20–27 years). Exclusion criteria were identical to those used in the main study; however, this time all participants qualified for the final analysis (minimal mean duration of exclusive percepts = 0.91 s, minimal frequency of alterations = 4.5, maximal cumulative duration of mixed percepts = 58.4%). The statistical analyses were identical to those from the main study.

## Study 2: Results and discussion

In study 2 –with the backward and forward optic flow speeds switched–we found that the main effect of congruence was just above the threshold of significance ($F(1, 11) = 4.41$, $p = 0.059$, $\eta^2_G = 0.030$), whereas neither the main effect of walking ($F(1, 11) = 0.19$, $p = 0.669$, $\eta^2_G < 0.001$) nor the interaction between both factors were significant ($F(1, 11) = 1.02$, $p = 0.334$, $\eta^2_G = 0.010$; S7A Fig). Pairwise comparisons showed no significant differences between conditions (all p values $> 0.283$). These results suggest that, even though the increased visual speed might enhance the access of the stimulus to awareness (see also [56–59]), the predominance of expanding optic flow in the main study cannot be accounted for solely by the difference in optic flow speeds.

Again, the mean self-reported level of overlap between button presses and dominant percepts was relatively high (M = 5.08 on a 7-point scale, SD = 0.79, range: 4–6) and there were no significant differences between walking conditions with respect to frequency of perceptual alterations (forward: M = 27.3, SD = 16.5; backward: M = 24.5, SD = 16.1, V(11) = 59, p = 0.126) or mixed percepts (forward: M = 39.8%, SD = 15.5%; backward: 40.7%, SD = 15.2%, t(11) = −0.44, p = 0.669). Notably though, the proportion of mixed percepts was noticeably higher than in the main study. Therefore, we decided to test whether the higher level of mixed percepts had obscured any (hypothetical) speed-driven effects. To do so, we re-analyzed the data from a sub-sample of twelve subjects from the main study (study 1), matched with respect to the distribution of mixed percepts observed in the present study (study 2: M = 40.3%, SD = 14.9%; resultant values from the sub-sample of study 1 derived using the optimal matching approach [60]: M = 40.7%, SD = 15.2%). The results mirrored closely the ones observed in the whole sample, with insignificant main effects of congruence ($F(1, 11) = 0.22$, $p = 0.645$, $\eta^2_G = 0.003$) and walking direction ($F(1, 11) = 2.39$, $p = 0.150$, $\eta^2_G = 0.020$), but a significant interaction between these factors ($F(1, 11) = 15.94$, $p = 0.002$, $\eta^2_G = 0.090$; S7B Fig). Bonferroni-corrected post hoc comparisons indicated the predominance of the congruent optic flow (M = 30.9%, SD = 10.3%) over the incongruent one (M = 26.0%, SD = 8.0%) while walking forwards (t(11) = 3.42, p = 0.022) and the opposite pattern for walking backwards (incongruent: M = 33.6%, SD = 8.4%; congruent: M = 28.1%, SD = 7.0%; t(11) = 3.82, p = 0.010). Again, no differences were found between walking conditions with respect to the duration of expanding (t(11) = −1.61, p = 0.772) or contracting optic flows (t(11) = 1.28, p = 1.000).

## General discussion

In this paper, we investigated whether the direction of one's locomotion affects perceptual awareness of visualisations of self-motion in space. In the task, participants walked forwards or backwards on a treadmill while each of their eyes was presented with a different optic flow

(expanding or contracting). We hypothesised that optic flows congruent with the direction of locomotion would predominate in visual awareness over incongruent ones. Moreover, this effect was expected to increase in proportion to the subject's precision of active proprioception, as assessed with a joint-position reproduction task. Contrary to our hypothesis, we did not find evidence for the impact of locomotion on perceptual access to optic flow patterns. Instead, our results indicate that expanding optic flow (i.e., what we observe when we walk forwards in everyday life) is prioritised in visual awareness independently of walking direction. Similarly, we found no evidence of a positive association between the reliability of active proprioception and the predominance of locomotion-congruent perceptions of self-motion.

Our results seem to be at odds with Bayesian accounts of perception according to which the brain uses probabilistic information to optimise inferences about the state of the environment [38,39,61]. Prior expectations–in the form of learned associations between multisensory signals–bias the interpretation of unimodal sensory input towards percepts congruent with information available across different senses. Such effects of signal relatedness have been convincingly demonstrated for binocular rivalry in the domain of exteroception [8–16]: for example, simultaneous auditory and tactile influences can sum up to boost the perceptual availability of an intermodally congruent visual image, but cancel each other out in the case of conflicting audio-tactile stimulation [11,62]. Given that, in everyday life, we are exposed to a certain repertoire of visuomotor associations (e.g., our visual field expands while moving forwards and contracts while moving backwards), one might expect similar congruence effects in the domain of action, namely increased perceptual awareness of stimuli representing the likely consequences of one's actions. However, the evidence from prior studies that used perceptual-suppression based paradigms is mixed. The null findings in the present study converge with previous research, which did not demonstrate enhanced visual awareness of locomotion-congruent optic flow patterns [31]. For rotatory hand movements, significant effects were found in a study by Maruya et al. [25], while other studies did not reveal increased perceptual access to stimuli rotating in congruence with manual actions [23,26,27; but see also 23 (study 2) for effects in a bistable motion paradigm that did not involve interocular suppression]. Whereas some of the discrepancies might be due to methodological differences between these studies [26, p. 8], the cumulative evidence seems to suggest that congruence effects due to bodily actions are either very context-sensitive or much less distinct than in the case of the widely-reported exteroceptive influences on visual awareness [8–16].

Moreover, we did not find evidence for our hypothesis–drawn from Bayesian accounts of perception–that the impact of kinaesthetic-proprioceptive information would be weighted by its relative precision. Individually-assessed reliability of active proprioception (approximated by the variance of errors in a limb-position reproduction task [45]) was not associated with increased effects of action on visual awareness. We might speculate that, given the negligible contribution of movement-related (proprioceptive-kinaesthetic) information to disambiguating the direction of visual motion, the precision of proprioceptive signals–even if it acts as a weighting factor–ends up being practically inconsequential. However, as the variance of errors on the active joint position reproduction task is likely to be determined by multiple (e.g., inter-individual) factors, it is perhaps too rough an approximation of proprioceptive precision. Whereas, for exteroceptive modalities, precisions can be more easily controlled through manipulation of signal properties (e.g., their level of noisiness [63]), it is still challenging to develop methods to assess proprioceptive (as well as interoceptive) signalling [43].

Speculatively, methodological factors could also have contributed to the lack of bodily movement effects on visual experience in the present study. Even though locomotion on a treadmill is felt as being much faster than on the ground [46,64,65], it still might have been that the action effects were hindered by the use of unnaturally slow walking speed. The fact

that the participants' hands held the side-rim of the treadmill could have additionally contributed to weakened overall feelings of locomotion. Despite the visualisations of self-motion being presented at subjectively realistic speed (including mid-peripheral vision) and coupled to rhythmic head movements while walking, the visual motion did not accelerate/decelerate according to the concurrent phase of the gait cycle. This constitutes another palpable difference between the current setup and multilevel cohesion of visuomotor signals under natural walking conditions. Perhaps the combination of such lesser-scale inconsistencies could have prevented the generalisation of learned (broader-level) sensorimotor associations to distinctively unfamiliar sensory conditions. These limitations could mostly be overcome with the use of a wireless VR headset in a space allowing for long periods of unidirectional walking. On the flip side, as thoroughly discussed by Paris et al. [31, p. 1192], multimodal effects on binocular rivalry have been demonstrated in studies utilising less realistic conditions, which seems to suggest that the issue of detectability of such effects cannot easily be reduced to the ecological validity of crossmodal (e.g., visuomotor) correspondences.

Alternatively, a more overarching speculative interpretation based on Cancellation theories of perception [66–68] could perhaps explain the scarcity of congruence effects in the domain of action influences on visual awareness [23,26,27,31, but see also 25]. According to this approach, the processing of self-generated sensory signals becomes suppressed in order to prioritise more informative (unexpected/uncontrolled) events. This theory is traditionally used to explain why self-administered tickling [66,69] or putting pressure on one's own body [70,71] are perceived as being less intense than when equivalent stimulation is produced externally. However, it can also be applied to other sensory domains, for example, to interpret the effects of reduced visual sensitivity to expected consequences of one's actions [72,73]. In line with this view, the influences of action-based predictions are suppressed as a redundant source of information about the state of the environment (see also [74]). Perhaps, this could explain why motor actions do not yield comparable congruence effects to those of exteroceptive stimulation or bodily signals that did not follow from voluntary movements–such as externally-produced whole-body rotations [18] or passive proprioceptive signalling of one's limb position [19]. However, this is not to say that, during voluntary movements, the influences of kinaesthetic-proprioceptive signals are generally suppressed in multisensory integration processes. For example, there is growing evidence for the widespread distribution of locomotion-related signals in the primary visual cortex (e.g., [75]; for reviews see [76–78]) and the modulation of the visually perceived self-motion speed by one's locomotor activity [79–82]. Notably, the processing of signals evoked by the stimulation of proprioceptive (muscular) nerves has been shown to be actually facilitated during voluntary movements (as opposed to cutaneous inputs which are then suppressed) [83]. More broadly, it might be suggested that while integration of proprioceptive-kinaesthetic and visual cues seems to play a limited role in disambiguating the perception of external (e.g., visual) signals, it primarily serves other (body-oriented and not necessarily perceptual) multimodal processes, such as action control [84,85], localisation of body parts [86,87], determination of body ownership [88,89] and peripersonal space [90,91].

Beyond the absence of locomotion effects on visual awareness, our results showed an overall dominance of expanding optic flow over contracting flow. Such effects have been found in previous binocular rivalry studies which did not involve bodily actions, but which did employ similarly structured (looming/expanding), though small-sized, visual stimuli. In all but one study [92], expanding random-dot patterns appeared to have greater access to visual awareness compared to analogous receding stimuli [93–95]. This perceptual preference for expansion was more pronounced when more naturalistic stimuli (such as concentric gratings or textures) were used [92,94]. These results are usually interpreted [92–94] in terms of prioritised processing of behaviorally urgent events [96], with sensory expansion/looming being a prime example

of an ecologically salient cue indicating potential danger or collision. In fact, neurophysiological evidence from humans and primates implies the existence of a dedicated network of subcortical and cortical areas that responds preferentially to objects moving towards the body [97,98]. This "looming interpretation"of previous studies seems particularly compelling given that their participants were exposed to visual expansion patterns (limited to a central visual field), which could not be attributed to self-movement, but rather to external causes. Notably though, when the perceived motion of dot patterns can be identified as a consequence of one's locomotion, the predominance of visual expansion was found to disappear, which suggests that active locomotion may veto the sense of looming and potential collision [31].

The literature also suggests an alternative explanation of the effects of perceptual preference for expansion [93–95], based on the fact that visual patterns related to self-locomotion are asymmetrically represented in the brain. There is strong neurophysiological evidence that the cells which respond to expansion outnumber the cells attuned to contraction in higher level visual motion areas, such as the ventral intraparietal (VIP) area [99,100] and the dorsal subdivision of the medial superior temporal (MSTd) area [101–103]. Moreover, the selective response of MSTd neurons to large-field motion patterns has been shown to not depend on the specific depiction of visual movement [104] (e.g., concentric rings, outlines of squares, random dots, etc). An asymmetry between expansion and contraction is not surprising given that our experience of optic flow mostly arises as a consequence of forward movement. The prioritised processing of expanding optic flow is likely to be coupled with an accumulation of locomotor experience, as reflected by the fact that visual evoked potentials do not differentiate between directions of optic flow in infants at the pre-locomotion stage (3–4 months), but do differentiate for those with experience of crawling or walking (11–12 months), indicating faster recruitment of the neuronal networks responsible for recognising forward (compared to backward) self-motion [105].

This asymmetry-based interpretation seems more suitable for explaining the effects observed in the present research. In contrast to previous binocular rivalry studies, the stimuli used in this study have the characteristics of naturally occurring optic flow–a wide-field visual shifting of the surrounding scene consistent with head movements and walking velocity. It is unlikely that such visualisations would be mistaken for looming objects, given the documented role of the VIP area in "parsing" visual information into self-motion and object-motion components [106] and our ability to account for self-movement to appropriately determine the velocity of elements in a scene [107]. This parsing process is most accurate when there is congruent visuo-vestibular stimulation, but it is robust even with limited or no vestibular input [108,109] (e.g., when travelling in a car at a constant speed). Interestingly though, perceptual preference for expansion was not found in a previous study on walking effects on binocular rivalry by Paris et al. [31], which used expanding/contracting random-dot patterns. Perhaps the wide-field immersive visualisations used in our research had a greater capacity to induce neural responses in higher visual motion areas (such as the MSTd and VIP regions) which are fine-tuned for preferential processing of forward optic flow [101–103]. It is worth noting that this seems consistent with the findings from stationary binocular rivalry studies which found a stronger preference for visual expansion when more naturalistic stimuli are used [92,94]. Nevertheless, these interpretations should be taken with caution, as only direct measurement of brain activity and/or nuanced disentanglement of possible contributing factors (such as the size of the visual field, spatial structure and detailedness of the stimuli, and the presence and accuracy of visuo-motor coupling) can clarify the mechanisms underlying the effects reported in the literature.

In conclusion, under the conditions used in this study, we did not find evidence for increased perceptual awareness of optic flows congruent with direction of walking. Our

findings suggest that kinaesthetic-proprioceptive processing might play a negligible role in clarifying the visually perceived direction of self-motion. Instead, the observed results indicate that visual processing might be tuned to visual flows consistent with our life-long experience of forward locomotion.

## Supporting information

**S1 Fig. Optic flow speed values chosen as being subjectively matched to velocity of walking (0.42 m/s).** Each dot's coordinates represent a participant's mean matched optic flow speed when walking backwards (x-axis) and walking forwards (y-axis). Coordinates of the blue cross represent the sample means from both conditions. The gray cross indicates the physically accurate optic flow speed for the walking pace used. The dashed line represents optic flow speeds being equal on both walking conditions. The results indicate an overall overestimation of matched optic flow speed (as compared to the locomotion velocity) in both conditions; however, this tendency is more pronounced in the forward-walking condition–the probability distribution of estimates shifts toward higher values for walking forwards than for walking backwards.
(TIFF)

**S2 Fig. Locomotion effects on perceptual awareness for optic flow patterns (unfiltered sample).** Expanding optic flow (congruent with forward locomotion and incongruent with backward movement) predominated visual awareness independently of walking direction. $^{**}$p < 0.01.
(TIFF)

**S3 Fig. Proprioceptive assessment results.** The positive correlations between performances for flexions and abductions in terms of (**a**) proprioceptive accuracy (mean proprioceptive error) and (**b**) proprioceptive precision (variance of proprioceptive errors). (**c**) A strong overall correlation between proprioceptive accuracy and proprioceptive precision.
(TIFF)

**S4 Fig. Intercorrelations between items of the SUS questionnaire assessing sense of presence in the VR environment.** $^{***}$ p < 0.001; $^{**}$ p < 0.01; $^{*}$ p < 0.05.
(TIFF)

**S5 Fig. Associations between self-reported sense of presence in the virtual environment and the contents of visual awareness.** (**a**) Sense of presence was negatively correlated with the proportion of mixed percepts, and (**b**) showed a tendency to be positively correlated with the degree of predominance of locomotion-congruent optic flows.
(TIFF)

**S6 Fig. Perceptual awareness of expanding optic flow while standing and walking in different directions.** No significant differences were found between training (standing) blocks and visually identical blocks with either forward or backward self-motion.
(TIF)

**S7 Fig. Effects of locomotion on perceptual awareness of optic flow patterns for both studies.** (**a**) Study 2: when contracting optic flow (congruent with backward locomotion) was faster than expanding flow (congruent with forward movement), no significant differences in cumulative durations of percepts were found in either walking condition (p values > 0.283). (**b**) Study 1 (sub-sample matched with respect to the distribution of mixed percepts in study 2): when expanding optic flow was faster than contracting flow, it dominated visual awareness

independently of walking direction (p values < 0.022). * p < 0.05; ns = non-significant. (TIFF)

**S1 Table. Shortened version of the Simulator Sickness Questionnaire (SSQ).** The subject is instructed: "Please rate the extent to which you are experiencing now each of the symptoms". (DOCX)

**S2 Table. Adapted version of the Slater-Usoh-Steed Questionnaire (SUS).**
(DOCX)

**S1 File.**
(DOCX)

# Acknowledgments

We thank Andrzej Nowak for helpful suggestions, and Julia Sudnik and Zuzanna Kozłowska for their assistance in data acquisition.

# Author Contributions

**Conceptualization:** Paweł Motyka.

**Data curation:** Paweł Motyka.

**Formal analysis:** Paweł Motyka.

**Funding acquisition:** Paweł Motyka.

**Investigation:** Paweł Motyka.

**Methodology:** Paweł Motyka, Mert Akbal, Piotr Litwin.

**Project administration:** Paweł Motyka.

**Resources:** Paweł Motyka.

**Software:** Paweł Motyka, Mert Akbal.

**Supervision:** Paweł Motyka.

**Validation:** Paweł Motyka, Piotr Litwin.

**Visualization:** Paweł Motyka.

**Writing – original draft:** Paweł Motyka.

**Writing – review & editing:** Paweł Motyka, Mert Akbal, Piotr Litwin.

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
