## [Decision Letter · Decision Letter 0]

6 Jan 2021

PONE-D-20-35242

Forward optic flow is prioritised in visual awareness independently of walking direction

PLOS ONE

Dear Dr. Motyka,

Thank you for submitting your manuscript to PLOS ONE. After careful consideration, we feel that it has merit but does not fully meet PLOS ONE’s publication criteria as it currently stands. Therefore, we invite you to submit a revised version of the manuscript that addresses the points raised during the review process.

As you will see below, your manuscript has been assessed by two expert reviewers. Please try to address all of their comments. Reviewer 2 suggests that additional data in a standing condition would be very useful to control for biases and to unravel possible small effects. I agree. On the other hand, the current pandemic might make further data collection difficult. Please see whether it would be possible for you to collect this data at present and, if not, please discuss this point as best as you can. 

We look forward to receiving your revised manuscript.

Kind regards,

Markus Lappe

Academic Editor

PLOS ONE

Journal Requirements:

2)  Please ensure that you have specified how you recruited participants to your study in the Methods section.

3)  We note that Figure 1 includes an image of a participant in the study. 

Reviewers' comments:

Reviewer's Responses to Questions

**Comments to the Author**

1. Is the manuscript technically sound, and do the data support the conclusions?

Reviewer #1: No

Reviewer #2: Partly

2. Has the statistical analysis been performed appropriately and rigorously? 

Reviewer #1: Yes

Reviewer #2: Yes

3. Have the authors made all data underlying the findings in their manuscript fully available?

Reviewer #1: Yes

Reviewer #2: Yes

4. Is the manuscript presented in an intelligible fashion and written in standard English?

Reviewer #1: Yes

Reviewer #2: Yes

5. Review Comments to the Author

Reviewer #1: Review PONE-D-20-35242

Forward optic flow is prioritised in visual awareness independently of walking direction

Motyka, Akbal, and Litwin

This is a follow-up on a very intriguing finding by Paris et al (2017), who had found, unexpectedly, that physical walking direction (backwards or forwards) does not bias competing patterns of optic flow.

The present study reproduces and extends upon this finding. With respect to Paris et al, authors have used a larger number of subjects. They also aimed to reduce the perceptual-motor mismatch between the flow stimulus and walking. For this they used treadmill walking rather than circular walking, and a structured flow pattern rather than a cloud of points. The results fully confirm the intriguing unexpected finding of Paris et al.

Over all, the paper is very well written, and provides a good overview of the literature. The methods are well described and the statistics are fine. Probably due to a misunderstanding of the instruction a large number of subjects had to be excluded but the supplementary results show that excluding these subjects this does not affect the conclusion.

There is one major issue, i.e., that the Discussion is too superficial and misses a number of important points. Although the authors do give an interesting but rather complicated and far-fetched explanation for the lack of an effect, they fail to discuss earlier explanations (e.g. by ref 31, Paris et 2017), as well as simpler explanations, as explained in detail below. Moreover, a conclusion is presented that is not supported by the data (see below).

Overall, the data are important and interesting, but a simple explanations should be favoured over complicated ones.

MAJOR

Discussion, second paragraph (line 515-535)

Line 515-516: This sentence is not clear. What is meant with “These results […]”? Apart from ref 31, the cited studies are not about locomotion, but rather on hand- (ref 23, 26) and eye movement (ref 27). The authors seem to suggest that action does not influence bistable percepts, but this cannot be so because in (31) it does make a difference if the subject sits or walks, and in (ref 23), there was an influence on perceived rotation direction. Moreover, the authors should refrain from claims about “most previous reports”, while citing just four studies.

Same paragraph, Line 528-530: “with one exception” is misleading. First: science is not a democratic business, and, even then, a majority of three to one can hardly be regarded as a convincing argument.

-Same sentence: “local”: this seems to be an arbitrary and ad-hoc category of movements.

Same paragraph, Line 533-535: states a conclusion:

“Therefore, the cumulative evidence suggests that – at least in interocular suppression-based paradigms – bodily actions do not favour an intermodally coherent interpretation of visual signals, or such effects are much less robust than in case of exteroceptive influences.”

This conclusion is not supported by the cited data. First: ref 25 cites an opposite finding (the authors seem to think that the finding is somehow invalidated by being “not recent”). Second, as note above, refs 23 and 31 also present examples of action that does influence the percept. Third, the authors cannot claim in a general sense that “intermodally coherent interpretation” is not favored in even for their own results because they ignore the “mixed percepts” even though these amount amount one fourth of the response time.

The authors do present an interesting explanation for their results on the basis of cancellation theory. It should be noted though, that cancellation theory describes a very different case, i.e., it is about sensorimotor consistence. Put in a simple way: as long as the self-stimulation is consistent with the predicted outcome, it does not lead to a prominent perceptual experience. However in the present results there is an inconsistency, and so it is not at all obvious that it can be explained with the cancellation theory.

Rather, the authors need to discuss the following aspects:

– The thesis/explanation of Paris et al. (ref 31) to explain the phenomenon should be discussed in the light of the present experiments: “Perhaps this reflects a coarser form of action/perception link than we originally envisioned”.

– The legs are little represented in the cortex (much less than hands, mouth, face), and so, there might little perceptual coupling for that reason.

– The asymmetry of forward-backward responses and the unusual occurrence of backward visual flow and backward propulsion (except rowers and reverse runners) might mean that the “backward” percept in fact was not the counterpart to the “forward” percept, but rather, that the “backward” responses were among the “mixed” percepts: the “backward” responses might thus rather represent a “contracting” percept, and the participants simply pressed the button for they would otherwise hardly ever press it (this might be consistent with Fig 3b lower panel, where the mixed and backward percept seem to alternate).

– Possibly the walking speed was too slow: especially at slow speeds, the central field will hardly contribute to a percept of self motion so that the lack of peripheral view is problematic in the present study.

– Since the hand held the side-rim of the treadmill, the subjects had a very strong perceptual reference of the stationary world, which might have additionally weakened the percept of self-motion. This would probably weaken a possible influence of walking direction.

– Virtually all people are subject to regular driving experience, and in this case, they receive a strong expanding flow in the central visual field without active movements. This might cause some extent of decoupling of locomotion and optical flow in the central field.

– The stimulus might still not be realistic enough / correspond well enough: It had no rhythmic lateral sway nor anteroposterior acceleration in phase with walking. (But see arguments against this by Paris et al.).

– It could be, that the displayed movement did not convey enough self-motion, i.e. was still too much expansion-contraction like: It presented no visual periphery, and did not follow the rhythmic sway of walking.

MINOR

Title: it would be good if “binocular rivalry” occurred in the title.

Line 89:

It is claimed that the structured environment is an improvement, but Paris et al (2017) did test with structured environments: “Moreover, in a supplemental experiment (Supplement 1), we replicated the main results using, instead of dots, textured wall and ceiling surfaces to present dissimilar perspective-defined optic flow patterns to the two eyes.”

Dissociation/decoupling due to lack of gait-associated lateral movements and anteroposterior accelerations in the display?

Strong dominance of forward-flow percept

Reviewer #2: This study by Motyka et al investigates the effect of walking direction on the perception of optic flow direction during binocular rivalry and its modulation by proprioceptive precision. Building up on the literature showing that signals from different sensory modalities can bias visual perception during binocular rivalry in favour of the visual stimulus congruent with the cross-modal one, the authors used a sophisticated and well controlled virtual reality setup in which binocular rivalry was generated between opposite optic flow directions (consistent with forward and backward walking direction) while participants walked on a treadmill either forward or backward. The main result of the study (in line with previous evidence) is that walking direction does not impact visual rivalrous perception, and that forward optic flow consistently dominates binocular rivalry independently of the walking direction. The authors interpret this null result in the framework of the cancellation theory.

The study is well conducted and controlled, the authors made a technical tour de force to design the experimental setup using VR and controlling visual stimulation velocity. The paper is well written and the relevant litterature is well discussed.

My main concern about the study is the lack of a baseline condition in which binocular rivalry between the two optic flow directions is measured while participants simply stand (I understand that they used this condition during the training, but they only recorded two 45-sec blocks per participant). I believe that measuring the intrinsic bias for the forward flow preference could be helpful to normalize for each participant the data recorded while walking to a baseline, visual-only, condition. The effect of walking direction (probably because of the cancellation theory) might be very small and concealed by the intrinsic visual bias, but without a baseline condition, the bias migh conceal the effect. Walking backwards is a highly unnatural condition, it is therefore possibe that the effect is present only for the forward direction, but it is impossible with the current data to disentangle the visual perceptual bias from the possible cross-modal effect.

An efficient way of testing this possibility and maximizing the effectiveness of cross-modal stimulation could be that of interleaving within an experimental block relatively short (eg. 20-30 sec) periods of walking and rest.

I understand that it might be very challenging in this period to perform an additional experiment, but I really think that it might be crucial to interpret the results correctly.

6. PLOS authors have the option to publish the peer review history of their article (what does this mean?). If published, this will include your full peer review and any attached files.

Reviewer #1: **Yes: **Marc H.E. de Lussanet

Reviewer #2: No

---

## [Author Response · Author response to Decision Letter 0]

24 Mar 2021

All responses have been included in the attached file ("Response to Reviewers").

---

## [Decision Letter · Decision Letter 1]

16 Apr 2021

Forward optic flow is prioritised in visual awareness independently of walking direction

PONE-D-20-35242R1

Dear Dr. Motyka,

We’re pleased to inform you that your manuscript has been judged scientifically suitable for publication and will be formally accepted for publication once it meets all outstanding technical requirements.

Kind regards,

Markus Lappe

Academic Editor

PLOS ONE

Additional Editor Comments (optional):

Reviewers' comments:

Reviewer's Responses to Questions

**Comments to the Author**

1. If the authors have adequately addressed your comments raised in a previous round of review and you feel that this manuscript is now acceptable for publication, you may indicate that here to bypass the “Comments to the Author” section, enter your conflict of interest statement in the “Confidential to Editor” section, and submit your "Accept" recommendation.

Reviewer #2: All comments have been addressed

2. Is the manuscript technically sound, and do the data support the conclusions?

Reviewer #2: Yes

3. Has the statistical analysis been performed appropriately and rigorously? 

Reviewer #2: Yes

4. Have the authors made all data underlying the findings in their manuscript fully available?

Reviewer #2: Yes

5. Is the manuscript presented in an intelligible fashion and written in standard English?

Reviewer #2: Yes

6. Review Comments to the Author

Reviewer #2: I would like to thank the authors for their revision of the manuscript.

I recognize that in these difficult pandemic times acquiring new data is very complicated. I appreciate the additional analyses performed by the authors as well as the revision of the manuscipt acknowledging its limitations and toning down the conclusions.

7. PLOS authors have the option to publish the peer review history of their article (what does this mean?). If published, this will include your full peer review and any attached files.

Reviewer #2: No

---

## [Editor Report · Acceptance letter]

21 Apr 2021

PONE-D-20-35242R1 

Forward optic flow is prioritised in visual awareness independently of walking direction 

Dear Dr. Motyka:

I'm pleased to inform you that your manuscript has been deemed suitable for publication in PLOS ONE. Congratulations! Your manuscript is now with our production department. 

Kind regards, 

on behalf of

Dr. Markus Lappe 

Academic Editor

PLOS ONE